# Telementoring versus face-to-face mentoring in the training of scleral fixation surgery of intraocular lenses

**Luiz Filipe Adami Lucatto**[1]*, **Juliana Moura Bastos Prazeres**[1], **Gabriel Castilho Sandoval Barbosa**[2], **Eduardo Amorim Novais**[1,3], **Ricardo Luz. Leitão Guerra**[4], **Emmerson Badaró**[1], **Luiz Henrique Soares Gonçalves de Lima**[1], **Eduardo Buchele Rodrigues**[1]

1 Ophthalmology Department, Federal University of São Paulo, São Paulo, São Paulo, Brazil, 2 Suel Abujamra Institute, Ophthalmology Department, São Paulo, São Paulo, Brazil, 3 Department of Ophthalmology, Centro Oftalmológico Città, Rio de Janeiro, Brazil, 4 Department of Ophthalmology, Leitão Guerra–Oftalmologia, Salvador, Brazil

* filipeadami@yahoo.com.br

## Abstract

### Purpose

To assess telementoring as a complementary tool for surgical training of a scleral fixation technique.

### Design

Randomized, controlled, two-arm, blinded clinical trial.

### Methods

Using a 3D visualization system, 132 participants were randomized in order of enrollment into traditional face-to-face mentoring (n = 66) and telementoring (n = 66). A single surgeon mentored all participants in the 2 groups. The procedure was performed in a model suitable for training in intraocular lens fixation SimulEYE® (INSEYET, WESTLAKE VILLAGE, CA, USA). In the telementoring group, the images captured on a local computer were sent to a second computer located in another room through a teleconferencing platform in real-time. Nine steps of the recorded procedure were evaluated and scored by two masked independent surgeons experienced in the technique.

### Main outcomes measures

The primary outcomes evaluated were the global score (the sum of each score on the rubric), surgical failure, and surgical time (in seconds).

### Results

Surgical success was achieved in 98.5% in the face-to-face group and in 95.5% in the telementoring group (p-value = 0.619). Minimal technical problems were reported in 8

**Data Availability Statement:** All relevant data are within the paper and its Supporting Information files.

**Funding:** This study received a grant for Research from Alcon®. The funders had no role in study design, data collection and analysis, decision to publish, or preparation of the manuscript.

**Competing interests:** The authors have declared that no competing interests exist.

procedures in the telementoring group (12%), without interfering with the surgical result, and completion of the procedure.

## Conclusions

Telementoring is an encouraging educational tool that can overcome geographical barriers to ease the transfer of abilities and knowledge. We lack evidence in terms of group differences for superiority comparing face-to-face and telementoring, in addition to presenting comparable results regarding surgical success and failure. More studies are needed to explore the impact of telementoring in other ophthalmological surgeries.

**Trial registration:** The Federal University of São Paulo institution's Research Ethics Committee reviewed and approved this study protocol (approval number, 5.383.484).

## Introduction

Surgical training is preponderantly associated with a face-to-face method of training in which the mentor and the student participate in the surgical procedure. This relationship is historically important in the surgeon's learning process during the training period. However, the constant evolution of technology in the medical field leads surgeons to deal with a greater complexity of surgical techniques and a greater volume of knowledge to be acquired during their careers. Thus, access to new technologies and the transfer of knowledge for new surgical techniques may face geographic barriers imposed on surgeons due to the limitation of being far from appropriate mentoring to learn these new procedures [1].

Another relevant limitation arose in the period of the pandemic related to COVID-19, where several residency programs worldwide had their elective surgeries postponed with no expected resumption. Continuing to teach courses, research projects, and scientific meetings also had to be discontinued, mainly harming the ability to learn for physicians in training [2–5]. A quick adaptation to this newly established scenario became necessary. The application of telemedicine is one modality that has the potential to help face these challenges. Since then, the growth of webinars, podcasts, and online symposia allowed remote interaction and the exchange of ideas.

Telemedicine is "the use of medical information exchanged from one site to another via electronic communications to improve a patient's clinical health status" [6]. Following that rationale, telementoring exerts a paramount role in increasing both quality and access to surgical care. In this modality, there is a relationship facilitated by telecommunication technology, in which a mentor provides guidance to a mentee from a remote location in real-time. It has been used in different ways, combining isolated audiovisual technologies, telestration (a tool that makes it possible to draw on the screen), robotic arms, and electrosurgical control, among others [7]. Previous studies have suggested that telementoring has a similar safety and efficacy profile as on-site mentoring in a variety of settings [8].

The growing use of 3-D visualization system technology in ophthalmology, especially in vitreoretinal surgeries, allows the surgeon to operate on patients s with excellent visual quality, minimal delay, and a good depth of focus. Systems, such as the NGENUITY® 3D Visualization System (Alcon, Fort Worth, TX, USA) allow surgeons to perform and monitor the procedure in real-time through the use of passive 3-D glasses with all experiencing the same view as the main surgeon. The use of this technology has been frequently used in congresses with live discussions of surgical procedures. Despite the growing use of this technology in ophthalmology, there is no

study in the literature that assesses the effectiveness of telementoring as an option to acquire new knowledge or improve surgical technique in ophthalmic surgeries.

The purpose of this study is to compare telementoring as a complementary tool for surgical training of a scleral fixation technique that uses a 4-haptic intraocular lens (Akreos A060 Bausch & Lomb, Rochester, NY) and Gore-Tex CV-8 polytetrafluoroethylene sutures (W.L. Gore & Associates, Newark, DE), with face-to-face mentoring via a 3D visualization system.

## Methods

### Study population

The Ethics Committee in Research of the Federal University of São Paulo, Brazil, approved this randomized, controlled, two-arm, blinded clinical trial (masked grader), which adhered to the tenets of the Helsinki Declaration. Retina and anterior segment surgeons, fellows and ophthalmology residents, with no previous experience with this particular technique of 4-point scleral lens fixation (Akreos® - Bausch & Lomb) using polytetrafluorethylene sutures (Gore-Tex® CV-8), were recruited, and informed about the research protocol, and each volunteer provided prospective written informed consent. The exclusion criteria were previous experience with this technique of scleral fixation.

One hundred and third two participants were selected and randomized in order of enrollment into traditional face-to-face mentoring (n = 66) or telementoring (n = 66). After consent was obtained, a single surgeon (LFAL) mentored all participants in the 2 groups. Demographic data collected in this study included: sex, age, career stage (residency, fellow, subspecialty), and time of career (in years) since the beginning of residency. The participants were also asked about previous experience with the NGENUITY® 3D Visualization System.

### Study protocol

Surgical training was done individually in a wet lab center in São Paulo, Salvador, and Recife (Brazil). Both groups had a training module explaining the technique before performing the procedure. The procedure was performed using a model suitable for training of intraocular lens fixation SimulEYE® (INSEYET, WESTLAKE VILLAGE, CA, USA). The NGENUITY® 3D Visualization System (Alcon, Fort Worth, TX, USA) was used to train all participants and was used for all procedures. The system videos were captured through a capture card (Cam Link 4K –Elgato), and the procedure videos were recorded without audio for analysis by the masked graders. In the face-to-face training group, the mentor was in the same room as the trainee during the whole procedure, directly explaining the steps of the surgery. In the telementoring group, the images captured on a local computer were sent via control protocol/ internet protocol (TCP/IP) transmission to a second computer located in another room. The images were transmitted through a videoconferencing platform (Zoom Inc., San Jose, California, USA) so that the mentor could follow the procedure in real-time. For communication between the surgeon in training and the mentor, another camera was used that captured the audio and images of the surgeon's face (Fig 1). Two independent surgeons (JMBP and EB), experienced in the technique, observed the videos in a masked fashion and scored the trainees' performance using a rubric containing nine surgical steps that were graduated on a 3 points Likert scale. Each surgical step has specific guidelines (S1 Table) that exemplify whether the surgeon "performed inappropriately or inefficiently"- score 1; "performed with some hesitation, with additional maneuvers, but in a satisfactory manner"- score 2, or "performed well and without hesitation, showing respect for technique, tissues, time and mobility"- score 3. The surgical steps evaluated were: A–"Marking of fixation points with a pen 180˚ apart"; B–"Marking the correct position of the 4 point sclerotomies (3mm from the limbus and 4-

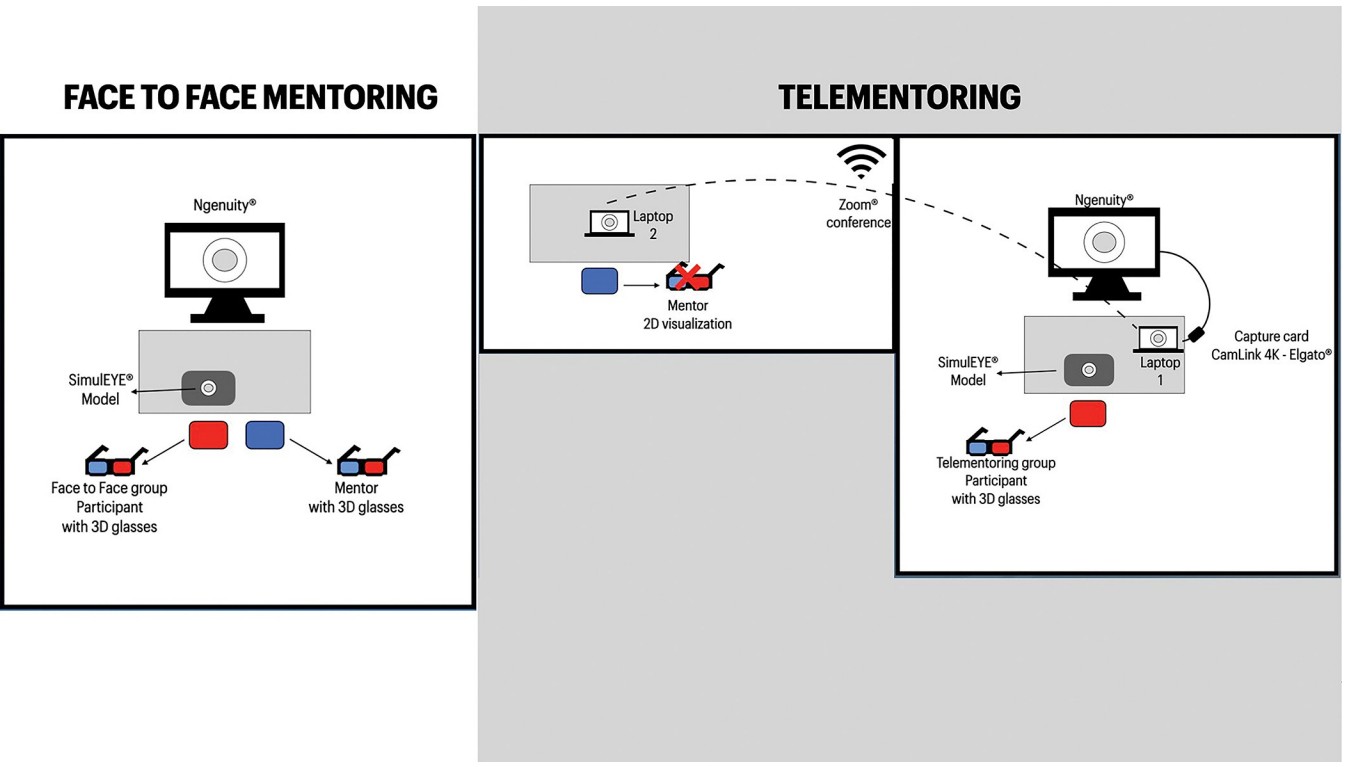

**Fig 1. Scheme showing the differences in training room layouts in the face-to-face and telementoring group.**

5mm apart) bilaterally"; C–"A 3-4mm corneal incision is made perpendicularly to the sclerotomies axis"; D–"GoreTex sutures is placed through sclera and corneal incision bilaterally, without crisscrossing the sutures"; E–"The sutures of the IOL haptics are made on the recommended orientation, without grasping the eyelets"; F–"The IOL is folded with a forceps and implanted into the anterior chamber"; G–"The IOL is centralized before starting the sutures"; H–"The sclerotomies are removed, and the sutures are made bilaterally without IOL decentralization"; I–"The GoreTEX sutures are buried into the sclera" (Fig 2).

At the end of the rubric, the evaluator was required to make a judgment about the surgical success of the procedure. The mentorship was classified as a Surgical failure if: "Failure in the transmission of images and/or audio, which makes it impossible to complete the mentoring"; "surgeon was not able to complete one of the surgical steps" and "the procedure had any surgical complication such as loss of suture, damage to the IOL, or displacement of the IOL into the vitreous cavity" (Fig 3).

The primary outcomes evaluated were the global score (the sum of each score on the rubric), surgical failure, and surgical time (in seconds).

At the end of the training, the participants answered a form to classify the quality of the mentorship on a 5-point Likert scale for: communication with the mentor; SimulEYE® model use; mentorship experience; NGENUITY® 3D Visualization System use. These quality parameters were the secondary outcomes measured in this study.

### Statistical protocol

**Sample size calculation.** The parameters used to estimate our sample size were power of 0.8, alpha error probability of 0.05, allocation ratio of 1:1, and an a priori difference effect on

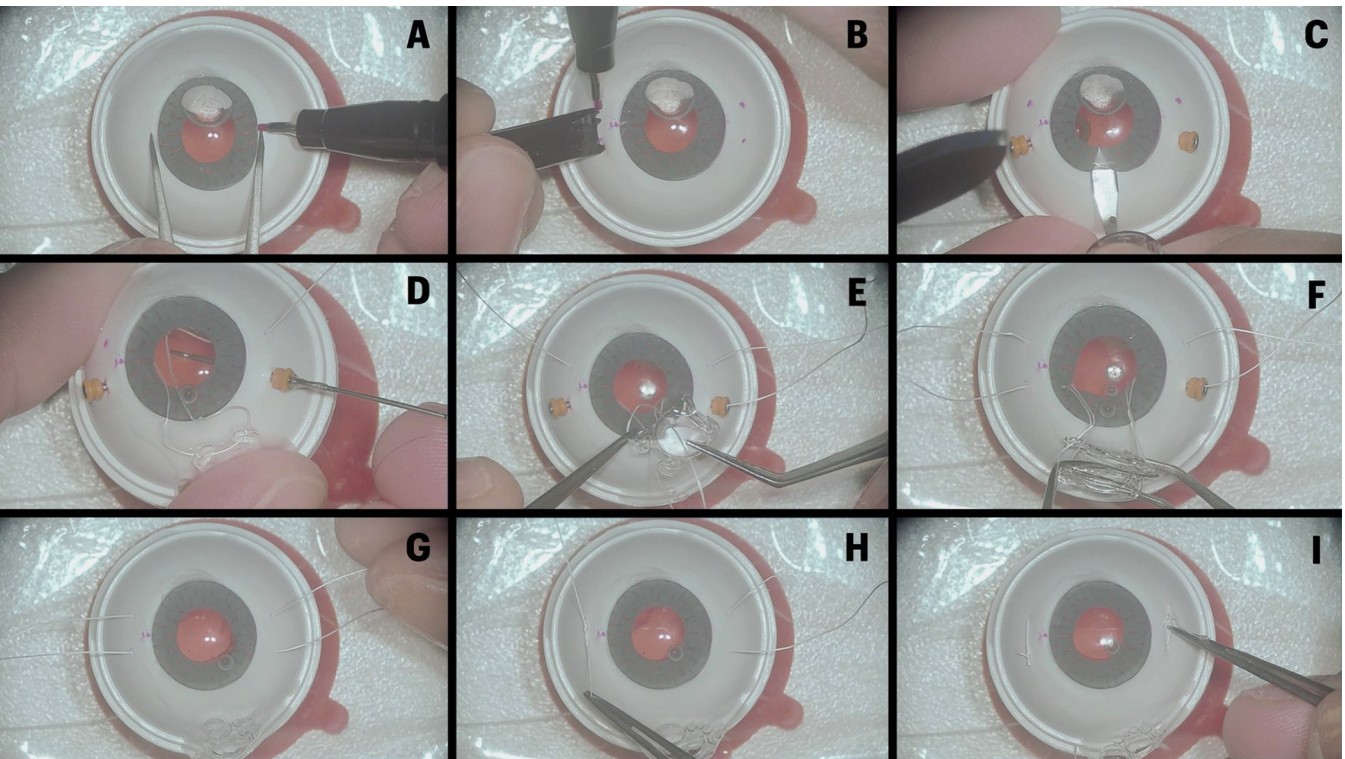

**Fig 2. The image illustrates the 9 evaluated steps of the global score.** Steps G and I were removed from the final scale. A–"Marking of fixation points with a pen 180° apart"; B–"Marking the correct position of the 4 point sclerotomies (3mm from the limbus and 4-5mm apart) bilaterally"; C–"A 3-4mm corneal incision is made perpendicularly to the sclerotomies axis"; D–"GoreTex sutures are placed through the sclera and corneal incision bilaterally, without crisscrossing the sutures"; E–"The sutures of the IOL haptics are made in the recommended orientation, without grasping the eyelets"; F–"The IOL is folded with a forceps and implanted into the anterior chamber"; G–"The IOL is centralized before starting the sutures"; H–"The sclerotomies are removed, and the sutures are made bilaterally without IOL decentralization"; I–"The GoreTEX sutures are buried into the sclera".

the score for skills for insertion of a four-point scleral fixation of an Akreos® IOL using Gore-TEX® sutures (scaled developed for this study) of a Cohen's d of 0.5 (a medium effect size). Under such input parameter, our sample size is 132 participants (66 vs. 66). Sample size was conducted using GPower [9].

**Statistical analysis.** Interrater agreement was calculated for the nine items of the scale for skills on this technique, and only those items showing % of agreement superior to 80% were preserved to be used in the final summed score.

T-tests were used to compare the mean difference between face-to-face and telementoring groups if the assumptions for parametric testing were fulfilled. Fisher exact tests were used to compare groups in terms of proportions. For t-tests, standardized effect sizes were calculated using Cohen's d, and the following cutoffs were considered: 0.2 as small, 0.5 as medium, and 0.8 as large [10].

Interaction effects for the primary outcome were used to explore the effects of telementoring depending on the years of career. In the case of an interaction effect, we might explore from which amount of years of experience the effects of the telementoring on the outcome might change via the Johnson-Neyman plot [11].

All analysis was carried out using SPSS version 24 and PROCESS macro [12]. for the interaction effects. The adopted significance level was 0.05.

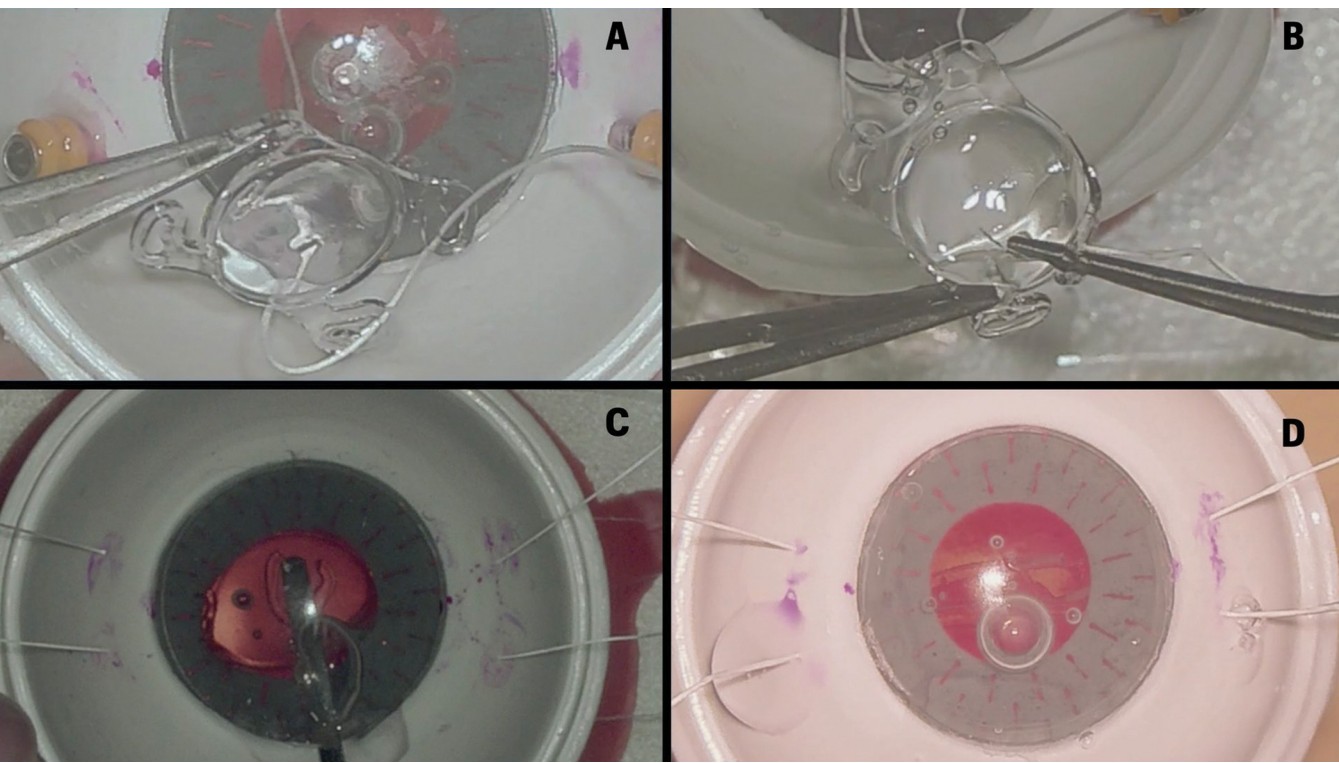

**Fig 3. Figure showing examples rated as incorrect by examiners.** A and B show a participant who scored 2 on the step "The sutures of the intraocular lens (IOL) haptics are made on the recommended orientation, without grasping the eyelets". In figure A, the participant held the IOL by the haptic more than once during mentoring. In figure B, the participant introduced the forceps inside the haptic and opened the instrument, generating stress in it. Figures C and D show participants who had mentoring judged as a failure. In C a torn Akreos® haptic and in figure D we see an Akreos® with a damaged optic portion.

## Results

The study included 73 (55%) men and 59 (45%) women with a mean age of 33.9 years (range 26 to 56 years). The mean career time was 7.19 years (range 0 to 30 years). Fifty-seven participants (43%) had never used the NGENUITY® system before the study. Most participants were specialists (44%), followed by Fellows (39%) and ophthalmology residents (17%). Table 1 shows the participants' demographic features and, as can be seen, the randomization created a balanced distribution between groups in terms of demographics such as age, career, sex, and previous use of NGENUITY®.

Table 2 shows the interrater agreement (all the items showed a percentage of agreement superior to 80%); however, the seventh and ninth items were excluded. The seventh step was performed correctly by all participants. The ninth step was excluded because burying the Gor-eTEX suture was not a feasibly reproducible step using the SimuEye® model. Statistically, these educational characteristics do not show enough variance to calculate the percentage of agreement for the seventh item and the non-significant p-value for the ninth. Therefore, the summed score (Global Score) across the seven remaining items ranges from 0 to 21 (Table 3). There is a lack of evidence between the groups (p-values > 0.05), and the standardized effect sizes were inferior to 0.2, indicating a very small effect under the Cohen's d-scale cutoffs. By lack of evidence, we mean that your findings refer to a situation where the analysis does not provide sufficient evidence to support the hypothesis (e.g., the difference between the two arms on the outcome had a p-value superior to the commonly adopted significance level [0.05]) [10].

**Table 1. Demographic features of the participants.**

| Nature of the variables | | Face-to-face (n = 66) | | | | Telementoring (n = 66) | | | |
|---|---|---|---|---|---|---|---|---|---|
| | | Minimum | Maximum | Mean | SD | Minimum | Maximum | Mean | SD |
| Continous | Age | 27 | 56 | 33,47 | 6,239 | 26 | 54 | 34,33 | 6,01 |
| | Time of carrer (years) | 0 | 30 | 6,92 | 6,52 | 1 | 29 | 7,45 | 6,01 |
| Categorical | Previous experience with Ngenuity® system | | | | | | | | |
| | Yes | 36 | 55% | | | 39 | 59% | | |
| | No | 30 | 45% | | | 27 | 41% | | |
| | Sex | | | | | | | | |
| | Female | 31 | 47% | | | 28 | 42% | | |
| | Male | 35 | 53% | | | 38 | 58% | | |
| | Carrer stage | | | | | | | | |
| | Ophthalmology residents | 12 | 18% | | | 11 | 17% | | |
| | Fellows | 26 | 39% | | | 25 | 38% | | |
| | Specialists | 28 | 42% | | | 30 | 45% | | |

Surgical success was achieved in 98.5% in the face-to-face group and in 95.5% in the telementoring group (p-value = 0.619). Only one participant from the face-to-face group had the procedure classified as "surgical failure" because of IOL damage during the Akreos® implantation. In the Telementoring group, two participants had surgical failure related to the same complication, and another participant due to a technical problem in the transmission of the images at the end of the mentorship.

Regarding secondary outcomes (mentoring satisfaction) where four items were evaluated, we observe that two of them were shown to be significantly positive and in favor of a face-to-face approach: communication and the SimulEye® model used for training, with effect sizes between small and medium (i.e., 0.2 to 0.5, standardized Cohen's d effect size).

Minimal technical problems were reported in 8 procedures in the telementoring group (12%), without interfering with the surgical result, and completion of the procedure. Six participants reported temporary interruptions in communication, and one complained of delay. Another participant reported a problem with the 3D NGENUITY® system and that the 3D visualization was not activated during mentoring.

**Table 2. Interrater agreement.**

| Surgical steps | Agreement | Expected agreement | Kappa | Std. Error | z | p-value |
|---|---|---|---|---|---|---|
| Marking of fixation points with a pen 180˚ apart | 90.15% | 77.89% | 0.5545 | 0.0828 | 6.70 | p<0.001 |
| Marking the correct position of the 4-point sclerotomies (3mm from the limbus and 4-5mm apart) bilaterally | 84.85% | 50.86% | 0.6917 | 0.0860 | 8.04 | p<0.001 |
| A 3-4mm corneal incision is made perpendicularly to the sclerotomies axis | 94.70% | 86.62% | 0.6038 | 0.0858 | 7.04 | p<0.001 |
| GoreTEX® sutures is placed through sclera and corneal incision bilaterally, without crisscrossing the sutures | 85.61% | 47.49% | 0.7259 | 0.0807 | 9.00 | p<0.001 |
| The sutures of the IOL haptics are made on the recommended orientation, without grasping the eyelets | 81.06% | 57.26% | 0.5569 | 0.0813 | 6.85 | p<0.001 |
| The IOL is folded with a forceps and implanted into the anterior chamber | 87.12% | 49.45% | 0.7452 | 0.0771 | 9.67 | p<0.001 |
| The IOL is centralized before starting the sutures | No variance | | | | | |
| The sclerotomies are removed, and the sutures are made bilaterally without IOL decentralization | 93.94% | 56.73% | 0.8599 | 0.0786 | 10.94 | p<0.001 |
| The GoreTEX® sutures are buried into the sclera | 99.24% | 99.24% | 0.0000 | 0.0000 | 0.00 | 0.5000 |

**Table 3. Primary and secondary outcomes.** The global score (seven items summed score) ranged from 0 to 21.

| | | Face-to-face (n = 66) | | | | Telementoring | (n = 66) | | | | |
|---|---|---|---|---|---|---|---|---|---|---|---|
| | | Minimum | Maximum | Mean | SD | Minimum | Maximum | Mean | SD | Effect size | p-value |
| Primary Outcome | Global score (seven items summed score) | 14,5 | 21 | 18,91 | 1,49 | 10 | 21 | 18.66 | 1,71 | 0,156 | 0,373 |
| | Surgery time (in seconds) | 1381 | 3817 | 2234,73 | 505,01 | 1510 | 3380 | 2209,36 | 510,442 | 0,05 | 0,775 |
| | | | | n | % | | | n | % | | |
| | Surgical Failure | | | 1,00 | 1,50 | | | 3,00 | 4,50 | | 0,619 |
| | | Minimum | Maximum | Mean | SD | Minimum | Maximum | Mean | SD | Effect size | p-value |
| Secondary Outcome | Communication quality | 4 | 5 | 4,94 | 0,24 | 3 | 5 | 4,77 | 0,49 | **0,432** | **0,014** |
| | SimulEYE® model use | 4 | 5 | 4,8 | 0,401 | 3 | 5 | 4,62 | 0,57 | **0,34** | **0,037** |
| | Mentoring experience | 4 | 5 | 4,95 | 0,21 | 4 | 5 | 4,94 | 0,24 | 0,067 | 0,7 |
| | Ngenuity® system use | 3 | 5 | 4,73 | 0,542 | 2 | 5 | 4,77 | 0,57 | 0,081 | 0,641 |

Lastly, as an exploratory analysis, we tested the effects of whether the primary outcomes might change depending on the time of experience since residency. We noted a significant interaction effect between the randomization assignment and time since the residency (p-value = 0.0457). Participants with more than 13 years since the residency within the telementoring group showed lower scores on the skill scale than those allocated in the face-to-face group. See Table 4 with the Johnson-Neyman table, where values for time since the residency

**Table 4. The influence of time since residency on the global score.**

| Time since the residency | Interaction effect | p-value | Lowe limit confidence interval | Upper Limit Confidence Interval |
|---|---|---|---|---|
| 0 | 0,4151 | 0,3187 | -0,4054 | 1,2357 |
| 1,5 | 0,2832 | 0,4425 | -0,4443 | 1,0107 |
| 3 | 0,1513 | 0,6444 | -0,4957 | 0,7983 |
| 4,5 | 0,0193 | 0,9479 | -0,5649 | 0,6036 |
| 6 | -0,1126 | 0,6837 | -0,6581 | 0,433 |
| 7,5 | -0,2445 | 0,3684 | -0,7806 | 0,2915 |
| 9 | -0,3765 | 0,1837 | -0,9337 | 0,1808 |
| 10,5 | -0,5084 | 0,0993 | -1,1143 | 0,0975 |
| 12 | -0,6403 | 0,0633 | -1,3165 | 0,0359 |
| **13,1307** | **-0,7398** | **0,05** | **-1,4796** | **0** |
| **13,5** | **-0,7723** | **0,0471** | **-1,5344** | **-0,0102** |
| **15** | **-0,9042** | **0,0393** | **-1,7632** | **-0,0452** |
| **16,5** | **-1,0361** | **0,0353** | **-1,9996** | **-0,0727** |
| **18** | **-1,1681** | **0,0332** | **-2,2415** | **-0,0946** |
| **19,5** | **-1,3** | **0,0321** | **-2,4873** | **-0,1127** |
| **21** | **-1,4319** | **0,0316** | **-2,736** | **-0,1279** |
| **22,5** | **-1,5639** | **0,0315** | **-2,9868** | **-0,1409** |
| **24** | **-1,6958** | **0,0316** | **-3,2394** | **-0,1522** |
| **25,5** | **-1,8277** | **0,0317** | **-3,4932** | **-0,1623** |
| **27** | **-1,9597** | **0,032** | **-3,7481** | **-0,1712** |
| **28,5** | **-2,0916** | **0,0323** | **-4,0039** | **-0,1794** |
| **30** | **-2,2235** | **0,0326** | **-4,2603** | **-0,1868** |

above 13 years are shown to have significantly lower scores (negative effects) among those subjects in the telementoring group. Regarding surgical time, we lacked evidence associating it with the interaction effect of career time (p-value = 0.41).

## Discussion

Telementoring is a promising tool that can overcome geographical barriers to ophthalmic education. This study reinforces the rationale that telementoring plays a teaching role similar to traditional face-to-face mentoring in surgical training of this scleral fixation technique using a 4-haptic intraocular lens and Gore-Tex CV-8 polytetrafluoroethylene sutures. We lack evidence in terms of group differences for superiority comparing face-to-face and telementoring, in addition to presenting comparable results regarding surgical success and failure. Considering surgeons with more than 13 years of experience since residency, there was a tendency for better results in the face-to-face group.

Telemedicine has seen astounding growth in recent years due to advances in key digital innovations in information and communication technology. Telementoring is an arm of telemedicine that uses data transmission technology to provide real-time supervision and technical assistance for surgical procedures from a specialist in a remote geographic location. It has advantages compared to traditional mentoring, including travel costs, distance, and availability.

Several studies in various medical specialties sought to compare the applicability of telementoring in clinical practice, comparing it with traditional face-to-face mentoring. In a systematic review conducted to evaluate the effectiveness of telementoring compared with on-site mentoring, Bilgic et al. [8] reported no difference in the complication rate in 11 studies. They reported similar operative times in 82% of the studies, and technical issues in 3% of all cases, also showing no difference in the acquisition of skills or knowledge between them.

In our study, 44% of the participants were retina and anterior segment specialists, 39% were fellows, and 17% were ophthalmology residents, however, the randomization created a balanced group in terms of demographics, such as age, career, sex, and previous use of the NGENUITY® system (57% of the participants had previously experienced the system). The primary outcomes evaluated were the global score (the sum of each score on the rubric), surgical failure, and surgical time (in seconds). As expected, we found that the overwhelming majority of our sample reached surgical success, with the success rate being similar between the two groups (98.5% in the face-to-face, and 95.5% in the telementoring), and this difference was not statistically significant (p-value = 0.619). The findings seemed to indicate that both mentoring techniques are effective, with no superiority of one over the other—the global score revealed no statistically significant difference (p-value = 0.373), with a mean score of 18.91 and 18.66 in the face-to-face and telementoring groups, respectively (Table 3). However, in the subgroup of participants with 13 or more years of experience since residency, the results in the telementoring group were inferior. Of the 132 participants in our sample, 22 had 13 or more years of experience. Inexperience with the Ngenuity® system is probably not the cause for these results since in our sample, 73% of this subgroup had previously used this technology.

The duration of the procedure also showed no statistically significant difference (p-value = 0.775), with a mean surgical time of 2.234,73 seconds (approximately 37 minutes and 15 seconds) in the face-to-face group, and 2.209,36 seconds (approximately 36 minutes and 49 seconds) in the telementoring group.

Regarding surgical failures, the face-to-face group reported one case (1.5%) whereas the telementoring group described three cases (4.5%). Nevertheless, the values obtained were not statistically significant (p-value = 0.619). The surgical failure from the face-to-face group

occurred because of IOL damage during the Akreos® implantation, and this also occurred in two cases for the telementoring group. The third case of surgical failure in the telementoring group occurred due to a technical problem in the transmission of images at the end of the mentoring period. In this case, the transmission image froze at the end of the procedure disabling the mentor to assist with the two last steps of the surgery (suture placement and burying of the suture points). This technical problem was related to the surgery recording software which ended unexpectedly and terminated the transmission.

with respect to the technical problems, we recorded a total of eight occurrences (6%) of which six (75%) were related to temporary interruptions in communication that resolved spontaneously and were probably related to network instability. Only one of the participants reported an episode of transmission delay, which did not influence the training outcome. Transmission delay is a very important concept that should be carefully examined when testing telementoring equipment. In our study, the mentor was located in a room next to where the mentee was conducting surgical training. Over long distances such as different cities, states, or countries, the delay can vary and exert a more relevant impact on telementoring. However, with advances in technology and the internet, as well as high-speed networking, these problems tend to be progressively reduced.

With regard to secondary outcomes, we obtained statistically significant values with a higher mean score in the "communication" (4.94 vs. 4.77; p-value = 0.014) and "SimulEYE®" use" (4.8 vs. 4.62; p-value = 0.037) items in the face-to-face mentoring group compared to telementoring group. The topics "mentoring experience" and "Ngenuity use" showed no statistically significant difference. We were not able to precisely substantiate the higher mean score in the topic "SimulEYE® use", since the item used by the participants had exactly the same characteristics. On the other hand, the "communication" item is understandable, since face-to-face mentoring does not present any communication challenges, while telementoring can be influenced by several issues such as volume, delay, and susceptibility to technical failure, among others.

Our study has limitations, mainly because it included only one surgical technique, about the results of which may or may not be extrapolated to other ophthalmological surgeries. In addition, telementoring with large distances between mentor and mentee was not evaluated, as both were only a room away. We also raise the limitations of reproducibility of telementoring implementation, which includes cost, equipment requirements, and legal and ethical issues. In addition, a pre-existing relationship between mentor and mentee is required (for example, through pre-telementoring sessions).

In this study, we used the Ngenuity® 3D Visualization System, which allowed the mentor to have the same 3D view that the main surgeon experienced, during face-to-face training. In the telementoring group, the mentor followed and assisted with the procedure in real-time with the images being transmitted through a teleconferencing platform in 2D. Another camera captured the audio and images of the surgeon's face for communication between the mentee and the mentor.

In order to fully implement telementoring in ophthalmology, updated versions of Ngenuity® could be developed by taking into account the possibility of real-time transmission of the surgeon's 3D visualization, audio transmission to allow different surgeons to interact simultaneously, even at great distances, with optimized reproducibility. That way, any viewer may experience the same view that the main surgeon is experiencing, with real-time transmission and 3D imaging, whether on-site or virtually.

Studies on telementoring in ophthalmology are limited. Camara et al. [13] guided a general ophthalmologist in the removal of a lateral orbital tumor from a site 210 miles away by using a 2-dimensiomal view and a videoconferencing system. Ye et al. [14] used one smartphone

adapted into a microscope with video sequences that were transferred in real time between two countries while the two parties conversed without any difficulty. Din et al. [15] recently published a study in which three surgeons in Toronto were proctored by a surgeon in Israel to implant a novel keratoprosthesis device into cadaver eyes. They also used NGENUITY® as well as a device to increase the transmission bandwidth.

To the best of our knowledge, this study is the first to compare face-to-face and telementoring groups in ophthalmic surgery. We chose the scleral fixation technique using a 4-haptic intraocular lens (Akreos A060 Baush & Lomb, Rochester, NY) and Gore-Tex CV-8 polytetrafluoroethylene sutures (W.L. Gore & Associates, Newark, DE) due to the fact that it is relatively recent, and most ophthalmologists still have no experience with it. In addition, it is a technique that presents good results with low complication rates.

In conclusion, this study demonstrates that telementoring is an encouraging educational tool that can overcome geographical barriers to ease the transfer of abilities and knowledge. More studies are needed to explore the impact of telementoring in other ophthalmological surgeries.

## Supporting information

**S1 Table. The evaluation form used in the study containing nine surgical steps graded on a 3-point Likert scale.**
(DOCX)

## Acknowledgments

We sincerely thank the participants of this study.

The Federal University of São Paulo institution's Research Ethics Committee reviewed and approved this study protocol (approval number, 5.383.484). The participants provided informed consent, and the confidentiality of the data collected during the survey is maintained by the authors.

## Author Contributions

**Conceptualization:** Luiz Filipe Adami Lucatto, Juliana Moura Bastos Prazeres, Eduardo Amorim Novais, Emmerson Badaró, Luiz Henrique Soares Gonçalves de Lima, Eduardo Buchele Rodrigues.

**Data curation:** Luiz Filipe Adami Lucatto, Juliana Moura Bastos Prazeres, Gabriel Castilho Sandoval Barbosa, Eduardo Amorim Novais, Ricardo Luz. Leitão Guerra, Emmerson Badaró, Luiz Henrique Soares Gonçalves de Lima, Eduardo Buchele Rodrigues.

**Formal analysis:** Luiz Filipe Adami Lucatto, Juliana Moura Bastos Prazeres, Gabriel Castilho Sandoval Barbosa, Eduardo Amorim Novais, Ricardo Luz. Leitão Guerra, Eduardo Buchele Rodrigues.

**Funding acquisition:** Luiz Filipe Adami Lucatto.

**Investigation:** Luiz Filipe Adami Lucatto, Juliana Moura Bastos Prazeres, Gabriel Castilho Sandoval Barbosa, Eduardo Amorim Novais, Emmerson Badaró, Eduardo Buchele Rodrigues.

**Methodology:** Luiz Filipe Adami Lucatto, Juliana Moura Bastos Prazeres, Gabriel Castilho Sandoval Barbosa, Emmerson Badaró, Luiz Henrique Soares Gonçalves de Lima, Eduardo Buchele Rodrigues.

**Project administration:** Luiz Filipe Adami Lucatto, Juliana Moura Bastos Prazeres, Gabriel Castilho Sandoval Barbosa, Eduardo Buchele Rodrigues.

**Resources:** Luiz Filipe Adami Lucatto, Juliana Moura Bastos Prazeres, Gabriel Castilho Sandoval Barbosa, Eduardo Amorim Novais, Ricardo Luz. Leitão Guerra, Luiz Henrique Soares Gonçalves de Lima, Eduardo Buchele Rodrigues.

**Software:** Luiz Filipe Adami Lucatto, Gabriel Castilho Sandoval Barbosa, Eduardo Amorim Novais, Ricardo Luz. Leitão Guerra, Eduardo Buchele Rodrigues.

**Supervision:** Luiz Filipe Adami Lucatto, Juliana Moura Bastos Prazeres, Gabriel Castilho Sandoval Barbosa, Emmerson Badaró, Eduardo Buchele Rodrigues.

**Validation:** Luiz Filipe Adami Lucatto, Juliana Moura Bastos Prazeres, Gabriel Castilho Sandoval Barbosa, Eduardo Amorim Novais, Emmerson Badaró, Luiz Henrique Soares Gonçalves de Lima, Eduardo Buchele Rodrigues.

**Visualization:** Luiz Filipe Adami Lucatto, Gabriel Castilho Sandoval Barbosa, Eduardo Amorim Novais, Ricardo Luz. Leitão Guerra, Luiz Henrique Soares Gonçalves de Lima, Eduardo Buchele Rodrigues.

**Writing – original draft:** Luiz Filipe Adami Lucatto, Juliana Moura Bastos Prazeres, Gabriel Castilho Sandoval Barbosa, Ricardo Luz. Leitão Guerra, Emmerson Badaró, Luiz Henrique Soares Gonçalves de Lima, Eduardo Buchele Rodrigues.

**Writing – review & editing:** Luiz Filipe Adami Lucatto, Juliana Moura Bastos Prazeres, Gabriel Castilho Sandoval Barbosa, Eduardo Amorim Novais, Ricardo Luz. Leitão Guerra, Emmerson Badaró, Luiz Henrique Soares Gonçalves de Lima, Eduardo Buchele Rodrigues.

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
