## [Decision Letter · Decision Letter 0]

10 Jul 2023

PONE-D-23-10630TELEMENTORING VERSUS FACE-TO-FACE MENTORING IN THE TRAINING OF SCLERAL FIXATION SURGERY OF INTRAOCULAR LENSESPLOS ONE

Dear Dr. Lucatto,

Thank you for submitting your manuscript to PLOS ONE. After careful consideration, we feel that it has merit but does not fully meet PLOS ONE’s publication criteria as it currently stands. Therefore, we invite you to submit a revised version of the manuscript that addresses the points raised during the review process.

We look forward to receiving your revised manuscript.

Kind regards,

Andrzej Grzybowski

Academic Editor

PLOS ONE

Journal Requirements:

"This study received a grant for Research from Alcon®."

3. We note that Figure 1 in your submission contain copyrighted images. All PLOS content is published under the Creative Commons Attribution License (CC BY 4.0), which means that the manuscript, images, and Supporting Information files will be freely available online, and any third party is permitted to access, download, copy, distribute, and use these materials in any way, even commercially, with proper attribution. For more information, see our copyright guidelines: http://journals.plos.org/plosone/s/licenses-and-copyright.

Reviewers' comments:

Reviewer's Responses to Questions

**Comments to the Author**

1. Is the manuscript technically sound, and do the data support the conclusions?

Reviewer #1: Yes

Reviewer #2: Yes

2. Has the statistical analysis been performed appropriately and rigorously? 

Reviewer #1: Yes

Reviewer #2: Yes

3. Have the authors made all data underlying the findings in their manuscript fully available?

Reviewer #1: Yes

Reviewer #2: No

4. Is the manuscript presented in an intelligible fashion and written in standard English?

Reviewer #1: Yes

Reviewer #2: Yes

5. Review Comments to the Author

Reviewer #1: Interesting and novel manuscript idea - the following comments are meant to strengthen:

-I am still unclear about the telementoring workflow - so was video being transmitted or only images? This is an important point to clarify

-Is career time post-training?

-Results section: please clarify this statement ' There is a lack of evidence between the groups (p-values > 0.05), and the standardized effects sizes were inferior to 0.2, indicating a very small effect under the Cohen’s d scale cutoffs.

-Given the surgical success scale used, is there any reference point to actual OR success? This would help contextualize the 98.5% and 95.5% values.

-If not inexperience with NGENUITY, why do you suppose did those with 13+ years of experience underperform wtih telemonitoring?

Reviewer #2: This study is interesting and has a relevant outcome for ophthalmic education. The analyzed group is sufficient and has been analyzed in an appropriate manner.

Specific remarks:

In Table 2, in the p-values row, "0.000" should be replaced with "p<0.001".

6. PLOS authors have the option to publish the peer review history of their article (what does this mean?). If published, this will include your full peer review and any attached files.

Reviewer #1: No

Reviewer #2: No

---

## [Author Response · Author response to Decision Letter 0]

18 Jul 2023

Journal Requirements: 

2. Please state what role the funders took in the study. 

3. We note that Figure 1 in your submission contain copyrighted images. We require you to either (1) present written permission from the copyright holder to publish these figures specifically under the CC BY 4.0 license, or (2) remove the figures from your submission:

We thank the Editorial Office and reviewers for the comments and suggestions. 

We have addressed the concerns raised, and we now believe that the manuscript is substantially improved. We hope that you will agree and find that the manuscript is now suitable for publication.

1) Ok. Done.

2) Ok. Done.

3) We have recreated an image to replace the previous "Figure 1", without copyrights.

Reviewer #1 

1) I am still unclear about the telementoring workflow - so was video being transmitted or only images? This is an important point to clarify

2) Is career time post-training?

3) Results section: please clarify this statement ' There is a lack of evidence between the groups (p-values > 0.05), and the standardized effects sizes were inferior to 0.2, indicating a very small effect under the Cohen’s d scale cutoffs.

4) Given the surgical success scale used, is there any reference point to actual OR success? This would help contextualize the 98.5% and 95.5% values.

5) If not inexperience with NGENUITY, why do you suppose did those with 13+ years of experience underperform wtih telemonitoring?

1) Yes, telementoring took place via videoconference. To avoid misinterpretation by readers, we have replaced the word "teleconferencing" with "videoconferencing" in the methods section.

2) We considered the baseline of the career time as the year in which the participant entered the medical residency in ophthalmology.

3) Here is written as “lack of evidence” because your findings refer to a situation where the analysis does not provide sufficient evidence to support the hypothesis (e.g., the difference between the two arms on the outcome had a p-value superior to the commonly adopted significance level [0.05]). It means that the observed results are inconclusive or do not provide strong support for a specific conclusion. It is common to observe that when RCTs describe group differences that are not statistically significant, authors mistakenly state that p-values > 0.05 as the intervention having no effect. No effect means evidence of absence which is different from lack of absence. The difference between lack of evidence and evidence of absence is classically described and explored in the statistical note written by Altman & Bland (1995). In our work, we had a special attention to clearly describing such statements. Now, in the manuscript, it might be found as follows on page 7 and 8, lines 170-175.

4) In fact, surgical success was considered if the participant managed to reach the end of the procedure without committing any criterion considered as failure. They are: “Failure in the transmission of images and/or audio, which makes it impossible to complete the mentoring”; “surgeon was not able to complete one of the surgical steps” and “the procedure had any surgical complication such as loss of suture, damage to the IOL, or displacement of the IOL into the vitreous cavity”. This information is already contained in the manuscript.

5) We do not know for sure, nor did we find clear reasons for this data obtained. At first, we thought that this subgroup possibly had less contact with the Ngenuity® system, but as mentioned in the manuscript, this did not happen: "Inexperience with the Ngenuity® system is probably not the cause for these results since in our sample, 73% of this subgroup had previously used this technology."

Reviewer #2 1) In Table 2, in the p-values row, "0.000" should be replaced with "p<0.001".

1) Ok. Done.

---

## [Decision Letter · Decision Letter 1]

1 Aug 2023

TELEMENTORING VERSUS FACE-TO-FACE MENTORING IN THE TRAINING OF SCLERAL FIXATION SURGERY OF INTRAOCULAR LENSES

PONE-D-23-10630R1

Dear Dr. Lucatto,

We’re pleased to inform you that your manuscript has been judged scientifically suitable for publication and will be formally accepted for publication once it meets all outstanding technical requirements.

Kind regards,

Andrzej Grzybowski

Academic Editor

PLOS ONE

**Comments to the Author**

1. If the authors have adequately addressed your comments raised in a previous round of review and you feel that this manuscript is now acceptable for publication, you may indicate that here to bypass the “Comments to the Author” section, enter your conflict of interest statement in the “Confidential to Editor” section, and submit your "Accept" recommendation.

Reviewer #1: All comments have been addressed

Reviewer #2: All comments have been addressed

2. Is the manuscript technically sound, and do the data support the conclusions?

Reviewer #1: Yes

Reviewer #2: Yes

3. Has the statistical analysis been performed appropriately and rigorously? 

Reviewer #1: Yes

Reviewer #2: Yes

4. Have the authors made all data underlying the findings in their manuscript fully available?

Reviewer #1: Yes

Reviewer #2: No

5. Is the manuscript presented in an intelligible fashion and written in standard English?

Reviewer #1: Yes

Reviewer #2: Yes

6. Review Comments to the Author

Reviewer #1: (No Response)

Reviewer #2: I have no further remarks. This study is interesting and has a relevant outcome for ophthalmic education.

7. PLOS authors have the option to publish the peer review history of their article (what does this mean?). If published, this will include your full peer review and any attached files.

Reviewer #1: No

Reviewer #2: No

---

## [Editor Report · Acceptance letter]

3 Aug 2023

PONE-D-23-10630R1 

TELEMENTORING VERSUS FACE-TO-FACE MENTORING IN THE TRAINING OF SCLERAL FIXATION SURGERY OF INTRAOCULAR LENSES 

Dear Dr. Lucatto:

I'm pleased to inform you that your manuscript has been deemed suitable for publication in PLOS ONE. Congratulations! Your manuscript is now with our production department. 

Kind regards, 

on behalf of

Dr. Andrzej Grzybowski 

Academic Editor

PLOS ONE